# Tellurium-Modified Nucleosides, Nucleotides, and Nucleic Acids with Potential Applications

**DOI:** 10.3390/molecules27238379

**Published:** 2022-12-01

**Authors:** Cen Chen, Zhen Huang

**Affiliations:** 1SeNA Research Institute and Szostak-CDHT Large Nucleic Acids Institute, Chengdu 610041, China; 2Firebird Biomolecular Sciences LLC, Alachua, FL 32615, USA; 3College of Life Sciences, Sichuan University, Chengdu 610064, China

**Keywords:** tellurium, crystallography, nucleic acids

## Abstract

Tellurium was successfully incorporated into proteins and applied to protein structure determination through X-ray crystallography. However, studies on tellurium modification of DNA and RNA are limited. This review highlights the recent development of Te-modified nucleosides, nucleotides, and nucleic acids, and summarizes the main synthetic approaches for the preparation of 5-PhTe, 2′-MeTe, and 2′-PhTe modifications. Those modifications are compatible with solid-phase synthesis and stable during Te-oligonucleotide purification. Moreover, the ideal electronic and atomic properties of tellurium for generating clear isomorphous signals give Te-modified DNA and RNA great potential applications in 3D crystal structure determination through X-ray diffraction. STM study also shows that Te-modified DNA has strong topographic and current peaks, which immediately suggests potential applications in nucleic acid direct imaging, nanomaterials, molecular electronics, and diagnostics. Theoretical studies indicate the potential application of Te-modified nucleosides in cancer therapy.

## 1. Introduction

Beyond all doubt, nucleic acids are the most important biomolecules for all known forms of life, which store all genetic information and pass it from one generation to the next. Extensive studies of nucleic acid have revealed its structure, functions, and properties [1,2,3,4]. The unique properties of nucleic acid promote its application in lots of areas such as diagnostics [5,6,7,8,9,10,11,12], therapeutics [13,14,15,16,17,18], crystallography [19,20,21,22], catalysis [23,24,25,26,27], material science [28,29,30,31,32], as well as vaccinology [33,34,35,36,37,38]. It is worth mentioning that nucleic acid as a therapeutic has greatly developed in recent decades and 14 oligonucleotide drugs have been approved since 1998, including siRNA, antisense, aptamer, etc. [39].

Notably, modified nucleic acid has played a vital role in almost all the above applications, because of the narrow chemical diversity and poor physiological stability of nucleic acids which restrict their utilization. Various strategies of modification have been employed to develop novel DNA and RNA analogs to overcome those limitations, such as changing the structure of the backbones [40,41,42,43,44,45,46,47,48], sugars [49,50,51,52,53,54,55], nucleobases [56,57,58,59,60,61,62,63,64,65], introducing different functional groups to endow the nucleic acid with specific function [66,67,68,69,70,71,72,73], or introducing special elements that do not exist in nucleic acids such as fluorine [74,75,76,77], bromine [78,79,80], mercury [81], and other heavy atoms [82].

As the same group element of tellurium, selenium modifications have been widely applied for protein structure determination, by replacing the sulfur in methionine, in which the selenium can be used as an ideal scattering center for multiwavelength anomalous dispersion (MAD) [83,84,85,86]. The selenium-derivatized nucleic acid (SeNA) has also achieved great success in 3D crystal structure determination and selenium has been introduced to different positions of the ribose, the phosphate backbone, as well as the nucleobases (Figure 1) [87,88,89,90,91,92,93,94,95,96,97,98]. These works have been well reviewed here [99,100]. It is worth mentioning that the incorporation of 2′-selenium modified nucleoside into DNA oligo not only solved the phase problem but also greatly facilitated crystallization, especially because, compared with protein crystallization, there are greater challenges in nucleic acid crystallization due to the negatively charged repetitive phosphate groups. Besides crystallography studies, selenium-modified nucleic acids have also been found to improve the specificity and sensitivity of DNA polymerization [12,47,101] However, compared with selenium, the application of tellurium in nucleic acid is still quite limited.

## 2. Discussion

Tellurium is a metalloid located in group VI, also known as chalcogens, of the periodic table following sulfur and selenium. Its chemical or physical properties, reactivities, structures, and functions have been well studied and show multiple potential applications in different areas [102,103,104,105,106,107]. Tellurium has a larger radius (1.35 Å) and a weaker electronegativity (2.0) in comparison to sulfur (1.04 Å, 2.58) and selenium (1.17Å, 2.55). The larger electrovalent and coordination sphere radius provide tellurium with strong metallic properties and result in weak covalent bonds with carbon and hydrogen [108]. Elemental tellurium has a lower abundance (1 ppb) than gold, platinum, or “rare-earth” elements on Earth [109]. Naturally occurring tellurium contains a series of isotopes, including ^120^Te (natural abundance 0.09%), ^122^Te (2.55%), ^123^Te (0.89%), ^124^Te (4.74%), ^125^Te (7.07%), ^126^Te (18.84%), ^128^Te (31.74%), and ^130^Te (34.08%) [110], which result in a unique isotope pattern in mass spectrometry for Te-containing compounds. Meanwhile, the diamagnetic nucleus ^125^Te (spin 1/2) enables Te-NMR studies and has wide chemical shifts ranging from −1400 ppm to 3400 ppm, which facilitates the identification of tellurium compounds with different chemical environments. Moreover, it also has excellent sensitivity due to the high natural abundance (7.07%) compared with ^13^C (1.1%) [111].

Although it belongs to the chalcogen family, the structures and chemical properties of tellurium compounds frequently differ from its family members [86,112]. For example, with the same range of oxidation states from −2 to +6 as sulfur and selenium, the higher oxidation states of tellurium are more stable due to the lower ionization energies [113,114]. In addition, the σ- and π- bond energies of tellurium are also significantly lower than their chalcogen analogs, [112,115,116,117] which contributes to its higher lability such as photochemical sensitivity [118] and lower tendency to catenate [119,120,121,122] compared to sulfur and selenium. Another intriguing feature of the chemistry of tellurium compounds is its proclivity to engage in hypervalent interactions, which was rationalized in terms of three-center–four-electron (3c–4e) bonding, charge-transfer interactions, hyperconjugation, and secondary bonding interactions (SBIs) [123,124,125,126,127,128,129].

Tellurium was successfully incorporated into protein in a tellurium-tolerant fungus in 1989, which was achieved by growing the Te-resistant fungi on a sulfur-free medium, and an extraordinarily high level of tellurium was detected [130]. Later, telluromethionine was reported to be selectively incorporated into dihydrofolate reductase [131]. Further studies optimized the bioincorporation technique of TeMet into protein and provide a promising approach for the X-ray structure study of protein [132]. The absorption edge of tellurium is about 0.3 Å, which indicates that it is not as suitable as selenium (0.9795 Å) as a scattering center in a MAD experiment. But the ideal electronic and atomic properties of tellurium for generating clear isomorphous signals make it a suitable heavy-atom for isomorphous replacement without the need for synchrotron radiation [108]. Interestingly, besides being covalently incorporated into protein to solve the phasing problem, the tellurium-centered Anderson–Evans polyoxotungstate (TEW) was used as a universal additive to enable or improve the crystallization of proteins to achieve high-quality crystals through electrostatic interactions [133]. Moreover, it was found that the incorporation of tellurium into phycocyanin (PC) and allophycocyanin (APC) enhanced their antioxidant activities [134].

### Te Modifications in Nucleoside, Nucleotide, and Nucleic Acids

The incorporation of Te into nucleic acid has also been achieved in the past decades. The first Te-modified nucleoside was reported [135] by Huang et al. in 2008 and then successfully incorporated into DNA oligo through solid-phase synthesis [136] (Figure 2). The tellurium functionality was protected by alkylation with either the phenyl or methyl group and was introduced into the 2′-position of both uridine and ribo-thymidine with good yield.

Interestingly, unlike the MeSe functionality [19,93,137,138,139,140], the 1′,2′- and 2′,3′-eliminations were observed during Te functionalization when using sodium borohydride as reducing reagent at room temperature (Figure 3), which provides a new method for the synthesis of the 2′,3′-didehydro-2′,3′-didoxynucleotides (d4Ns).

To get the desired product X, a stronger reducing reagent and lower temperature (0 °C) were applied together with crown ether (12-*crown*-4) to chelate the lithium ions to enhance the MeTe reactivity. The desired product was obtained in 47% yield with the 1′,2′-elimination products as the major byproduct. Both PhTe- and MeTe-modified nucleosides were incorporated into DNA oligos by solid-phase synthesis following a standard protocol [141] and quantitative coupling yield was achieved. A few of Te-DNAs have been oxidized to tellurides during the solid-phase synthesis which can be reduced by treating with diborane after the deprotection step (Figure 4). It was found that both methyltelluride and phenyltelluride functionalities were stable with the treatment of mild acid and base during the deprotection and purification. Interestingly, under heating (50 °C) in the presence of B_2_H_6_ or I_2_, 2′-TePh DNA undergoes 2′,3′-elimination at the modification site and generates the fragmented product. However, the 1′,2′-elimination was observed for 2′-TeMe DNA and creates the abasic product (Figure 4). The decrease in the melting temperature was observed during the UV melting study which was probably caused by the perturbation introduced by the bulky Te functionality (Figure 1).

In 2011, 5-PhTe modified nucleoside was successfully synthesized by applying the lithium–halogen exchange reaction [142] on a protected 5-iodo-2′-deoxyuridine and achieved medium yield (64%) [143] (Figure 5). The key steps of the reaction are the deprotonation of the NH and the treatment with n-BuLi followed by the addition of Ph_2_Te_2_. An elevated concentration of the reactant (0.15–0.18 M) is necessary to avoid the generation of a 6-PhTe isomer, which is inseparable. The synthesis of the corresponding phosphoramidite followed the standard protocol and applied to solid-phase synthesis. The results show that the PhTe functionality is well compatible with the solid-phase synthesis condition, deprotection, and purification. The UV-thermal denaturation studies show similar stability between the Te-derivatized duplex and the corresponding native which suggested that the bulky PhTe moiety is well accommodated and does not significantly change the duplex structure (Figure 2).

The Te-DNA crystal structure was also obtained by the same author by using 2′-Se modification strategy [137,138,139,144]. The results reveal that Te-DNA has virtually identical global and local structures as the corresponding native DNA (Figure 3). This result further confirms that the Te-functionality does not cause significant perturbation. The Te-DNA was quite stable under a high temperature (90 °C) but it was found to be sensitive to X-ray irradiation. Partial cleavage of the Te–C bond was detected through MALDI-TOF-MS after X-ray irradiation. Due to the metallic property of the tellurium atom, STM imaging studies show stronger topographic and current peaks for the Te-modified DNA duplex compared to the native one (Figure 4).

Photodynamic therapy (PDT) is a promising medical treatment using visible light irradiation in conjunction with a photosensitizer (PS), which is non-toxic in the dark, to selectively treat the targeted issue [144,145]. A study [146] investigated the photophysical properties of 2-/4-position Te-substituted thymidine, indicating its potential application as a UVA chemotherapeutic agent. The lowest triplet states were found to lie above the energy required to produce cytotoxic excited oxygen molecule and the absorption energies are short enough to penetrate the issue, via density functional theory (DFT) and time-dependent density functional theory (TDDFT) calculation. Further study [147] of the Te-subsituted deoxyguanosine revealed its ability to act as photosensitizer in cancer therapy.

## 3. Conclusions

The Te-modified DNA and RNA are a promising strategy for investigating the structure and function of nucleic acid. However, studies in this area are still quite limited and only a few papers on the subject have been published in the last 10 years. The 2′- and 5-position tellurium modified nucleoside has been successfully synthesized, and both are compatible with solid-phase synthesis, deprotection, and purification. The particular redox properties and selective elimination of the 2′-Te modified DNA oligo could be useful in studying DNA fragmentation and nucleobase damage. The location of the Te functionality modification and the size of the protecting group directly affect the melting temperature of the duplex, which could be used as a useful strategy for detecting DNA and RNA polymerization and catalysis. Furthermore, due to the metallic property of the tellurium atom, the Te-modified DNA duplex becomes visible under STM, which suggests a promising strategy for the direct imaging of DNA without structural perturbation. This will further help us conduct mechanism and function studies or even produce novel nano-electronic materials.

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
