# Peer review of "Tellurium-Modified Nucleosides, Nucleotides, and Nucleic Acids with Potential Applications"

_molecules, 2022, doi:10.3390/molecules27238379_

Round 1

Reviewer 1 Report

The review paper “Tellurium-Modified Nucleosides, Nucleotides and Nucleic Acids with Potential Applications” is significant for readers. It's suggested this paper to be published in Molecules after minor revision as labelled in the attached PDF file.

Author Response

Dear reviewer,

We appreciate you taking the time to review our paper. All changes were made according to the marks in the draft. The revised manuscript was attached.

Best,

Cen

Reviewer 2 Report

The heavy element Tellurium has been incorporated into proteins and used to determine the structure of proteins using X-ray crystallography. The review highlights the development of Te-modified nucleosides, nucleotides, and nucleic acids. A tellurium-modified 2' and 5-position nucleoside has been successfully synthesized and both are compatible with solid phase synthesis, deprotection and purification. Tellurium's ideal electronic and atomic properties for generating clear isomorphic signals give Te-modified DNA and RNA great potential for determining three-dimensional crystal structure using X-ray diffraction. The STM study also shows that Te-modified DNA has strong topographic and current peaks, immediately suggesting potential applications in direct nucleic acid imaging, nanomaterials, molecular electronics, and diagnostics. The special redox properties and selective removal of the modified 2'-Te DNA oligonucleotide can be useful in the study of DNA fragmentation and damage to nitrogenous bases. In addition, due to the metallic properties of the tellurium atom, the Te-modified DNA duplex becomes visible on STM, suggesting a promising strategy for direct DNA imaging without structural disruption. Theoretical studies point to the potential use of Te-modified nucleosides in cancer therapy.

The article is undoubtedly interesting and can be published after “minor revision

Author Response

Dear reviewer,

We appreciate you taking the time to review our paper. Minor changes about spelling and style were made. The revised manuscript was attached.

Best,

Cen
